# Detecting Metabolic Thresholds from Nonlinear Analysis of Heart Rate Time Series: A Review

**DOI:** 10.3390/ijerph191912719

**Published:** 2022-10-05

**Authors:** Giovanna Zimatore, Maria Chiara Gallotta, Matteo Campanella, Piotr H. Skarzynski, Giuseppe Maulucci, Cassandra Serantoni, Marco De Spirito, Davide Curzi, Laura Guidetti, Carlo Baldari, Stavros Hatzopoulos

**Affiliations:** 1Department of Theoretical and Applied Sciences, eCampus University, 22060 Novedrate, Italy; 2IMM-CNR, 40129 Bologna, Italy; 3Department of Physiology and Pharmacology “Vittorio Erspamer”, Sapienza University of Rome, 00185 Roma, Italy; 4Department of Teleaudiology and Screening, World Hearing Center, Institute of Physiology and Pathology of Hearing, 02-042 Warsaw, Poland; 5Heart Failure and Cardiac Rehabilitation Department, Faculty of Medicine, Medical University of Warsaw, 03-042 Warsaw, Poland; 6Institute of Sensory Organs, 05-830 Warsaw, Poland; 7Fondazione Policlinico Universitario A. Gemelli IRCCS, 00168 Rome, Italy; 8Neuroscience Department, Biophysics Section, Università Cattolica del Sacro Cuore, 00168 Rome, Italy; 9Department Unicusano, Niccolò Cusano University, 00166 Rome, Italy; 10Clinic of Audiology & ENT, University of Ferrara, 44121 Ferrara, Italy

**Keywords:** metabolic threshold, heart rate variability, sport, recurrence quantification analysis, nonlinear dynamic, Poincaré plot, wearable devices

## Abstract

Heart rate time series are widely used to characterize physiological states and athletic performance. Among the main indicators of metabolic and physiological states, the detection of metabolic thresholds is an important tool in establishing training protocols in both sport and clinical fields. This paper reviews the most common methods, applied to heart rate (HR) time series, aiming to detect metabolic thresholds. These methodologies have been largely used to assess energy metabolism and to identify the appropriate intensity of physical exercise which can reduce body weight and improve physical fitness. Specifically, we focused on the main nonlinear signal evaluation methods using HR to identify metabolic thresholds with the purpose of identifying a method which can represent a useful tool for the real-time settings of wearable devices in sport activities. While the advantages and disadvantages of each method, and the possible applications, are presented, this review confirms that the nonlinear analysis of HR time series represents a solid, robust and noninvasive approach to assess metabolic thresholds.

## 1. Introduction

During incremental intensity exercises, metabolic changes are observed and therefore the identification of metabolic thresholds allows personalized workload prescription milestones to be assessed. According to the Skinner diagram [1], during increasing intensity exercise, three phases of the body’s energy usage and two threshold points might be established.

With increasing exercise intensity, the aerobic threshold (AerT) [1] describes the transition from aerobic metabolism (production of energy using oxygen) to anaerobic metabolism (production of energy without oxygen); the anaerobic threshold (AnT) [2] describes the transition in which an over-proportional increase in ventilation occurs as a result of increased carbon dioxide production. During incremental exercise, these thresholds suggest two different workloads. The first threshold (AerT) defines the upper limit of an almost exclusive aerobic exercise with a low blood lactate concentration (about 2 mM/l). The threshold AnT is related to the highest workload of maximal lactate production rate in a steady state, i.e., when the production and the removal of blood lactate concentration are in equilibrium (about 4 mM/l). From a practical point of view, the AerT detection is widely used both in medicine and in the sports fields, where the threshold detection is used for investigating sedentary subjects and the active recovery processes. On the contrary, the AnT detection concerns almost exclusively amateur and professional athletes, and its role is to create targeted training programs [3,4].

These thresholds are usually detected by ventilatory (gas exchange) or metabolic (blood lactate) parameters [2]. The gas exchange testing (cardiopulmonary exercise test, CPET) and the blood lactate sampling are the gold standards in threshold detection [5,6]. Threshold estimation occurs by assessing heart and lungs performance at rest and during exercise [7], or evaluating blood lactate concentration [4]. Despite their efficiency, the widespread use of these methods is severely impaired due to them being highly invasive (as they require blood draws) and impractical (as they require a laboratory equipped with specific devices and trained operators) [8].

Several papers have demonstrated the validity of analysis of heart rate temporal series during sports activities [9,10,11,12,13,14,15,16,17,18,19,20,21,22] (see Table A1 of Appendix A). This is an important advancement in the field allowing many of the limitations already reported to be overcome. Indeed, although HR is commonly detected by using a conventional ECG device that requires the use of on-body electrodes wired to a recorder [23], nowadays, a number of wearable smart devices (wrist, watch [24], phone [25], finger ring [26], etc.) are available, allowing the measurement of HR quickly, continuously and in all situations of everyday life including fitness activities. These devices optically detect HR trough measurements of volumetric changes of blood in a vein, usually located at the level of the wrist (photoplethysmography—PPG) [27]. The detection accuracy of some of these devices [28,29], that can also contextually detect position, velocity and acceleration, has been certified by the Food and Drug Administration (FDA).

Despite the undoubted advantage in using HR time series, we should note that, typically, data analysis techniques are based on the assumption of signal stationarity. However, the inherently nonstationary nature of HR [30], which physiologically changes continuously to adapt to external stimuli, may generate unreliable conclusions. Although several signal preprocessing methodologies have been proposed to overcome such issues [9,11,30,31,32,33,34], approaches based on nonlinear analysis appear to be the more common approach.

This review is structured as follows: in Section 2, the main characteristics of HR time series analysis are described. In Section 3, a comparison is presented among the most representative nonlinear methods, such as the Poincaré geometry [21], the Detrended Fluctuation Analysis (DFA) [16,35], the Entropy [36,37,38] and the Recurrence Quantification Analysis (RQA) [30,33]. In Section 4, the applications of these techniques in the detection of metabolic thresholds are reported. In Section 5, the advantages and limitations of these methods are presented. Finally, in the Appendix A there is a description of the statistical procedures used in each section.

## 2. Heart Rate Time Series Analysis

Heart physiology is one of the first areas in Biology where the ideas of complexity and deterministic chaos (familiar to studies in Physics) produced practical applications [31]. As previously mentioned, nonlinear methods of HR classification have been developed to highlight and emphasize HR nonlinear fluctuations.

The oscillation patterns of a healthy heart are complex and constantly changing, allowing the cardiovascular system to rapidly adjust the sudden physical and psychological challenges to homeostasis [30]. The autonomic nervous system (ANS) controls HR through the balancing action of the sympathetic (S) and the parasympathetic (PS) nervous system. Increased S or diminished PS activity results in cardiac acceleration. Conversely, a low S activity or a high PS activity causes cardiac deceleration [23,39]. An additional fine HR regulation is maintained by the respiratory control centers which modulate the vagal outflow in the brainstem [10,31,40]. In this context, HR and heart rate variability (HRV) are strictly regulated by ANS and can provide relevant details on its dynamics and control mechanisms. To analyze HR signals, feature-based time-domain methods, frequency-domain methods and non-linear time-domain methods are typically used.

### 2.1. Feature-Based Time-Domain Methods

The methods based on the analysis of the time evolution of an ECG signal or a beat-by-beat HR record (as a discrete time series, Figure 1) are termed as time-domain analysis methods.

HR record can be expressed in bpm (beats/minute), but the main descriptors of the time-domain analysis are expressed in the literature in terms of the RR interval (milliseconds, ms). RR is the distance between two successive R waves of the QRS signal on the electrocardiogram. Time-domain features are denominated as [23,40]:RMSSD, the root mean square of difference between adjacent RR intervalsSDNN, the standard deviation of all RR intervalsSDSD or SDDS, the standard deviation of differences between adjacent RR intervals (or RR series)MSD, the mean successive difference between adjacent RR intervalsMASD, the mean absolute successive difference between adjacent RR intervals

### 2.2. Frequency-Domain Methods

The power spectrum analysis techniques transform time series data into frequency-domain data. The fast Fourier transform (FFT) and the short time Fourier transform (STFT) are the most common power-spectrum analysis techniques [41,42]. These spectral methods require a mathematical manipulation of the data and large time series (i.e., large amounts of data points). Conventional analysis methods based on the Fourier transform technique are not very suitable for the analysis of non-stationary signals, since these signals change over time. The main principle of STFT consists in choosing a small enough window for analysis in which the signal can be considered stationary. Moreover, when slowly and rapidly changing transient events occur, the STFT introduces significant errors. As the analysis window of the time-domain data gets narrower, the time resolution and the assumption of stationarity is improved, but the frequency resolution is reduced.

Frequency-domain analysis can distinguish between high frequency (HF) and low frequency (LF) components in the data and can identify peaks related to autonomic control [43] (see Figure 2, the peaks in green and blue). Since the HF and the LF correspond to short and long-latency components in the HR time series, they can be, respectively, considered as qualitative indicators of sympathetic and parasympathetic activity [39].

During heavy exercise (i.e., work intensity above AerT), there is a prevalence of HF components in contrast to LF components. The opposite effect has been observed during moderate exercise, i.e., prevalence of LF components compared to HF [18]. Unfortunately, during long-term recordings, there is a lower HR modulation stability which introduces significant difficulties in the interpretation of the frequency-domain analysis data. For this reason, time-domain methods are primarily used for data analysis.

To overcome the lower HR modulation stability, Cottin et al. (2004) [18] proposed a method where the mean high frequency peak (HF_peak_, Hz) and the area of HF power (HF_p_, ms^2^) could be computed automatically from the spectrogram. Then, the HF product [18,44] can be estimated by multiplying the HF_peak_ (Hz) by HF_p_ (ms^2^).

STFT and a time–frequency domain method were applied in the detection of the metabolic thresholds, and their applications are discussed in Section 4.

## 3. Nonlinear Analysis for HR Time Series

### 3.1. Poincarè Geometry

The Poincaré plot is a geometrical and nonlinear analysis method with numerous applications in different scientific domains [45], and it is extensively used for a qualitative visualization of the physiological signals involved [12,13,14]. It represents one of the most used methods in the assessment of heart rate dynamics in recent literature. In the review by Henriques et al. (2020) [34], it was reported that over 300 publications involving this analysis method on HR time series have been published, resulting in over 18,000 citations.

In the Poincaré plot, all values of a time series are plotted against previous values, leading to an ellipsoidal point cloud (Figure 3). Plot shapes can be categorized into functional classes to indicate cardiac functioning [13,15]. Moreover, this diagram may be analyzed quantitatively by the standard deviations of the distances *RR–RR(i)* to the lines y = x and y = −x + 2(*RR–RR_m_*), where *RR–RR_m_* is the mean of all *RR–RR(i).* The standard deviations are referred to as SD1 (green line) and SD2 (blue line) (Figure 3). Starting with the relationship between successive beats and the behavior of the heart, the graphic gives summary relevant information: the transverse axis of the ellipse (SD1) is a measure of the short-term changes in the RR intervals and is considered as an indicator of the parasympathetic activity; the longitudinal axis (SD2) reflects the long-term changes in the RR intervals, and it is considered as an inverse indicator of the sympathetic activity.

The correlations between SD1 and SD2 are enclosed into two important indices [12,31]: the sympathetic stress index (stress score = 1000/SD2) and the sympathetic–parasympathetic ratio (S/PS ratio = SD1/SD2), which have been recently described by Orellana et al. (2015) [23] in a longitudinal cohort study on soccer players.

One of the first Poincaré studies on HR (Tulppo et al. (1996) [12]) showed how this approach could provide useful relevant information of the HR dynamics during exercise, for events which are not easily detected by the conventional feature-based methods, such as during atropine administration and after a parasympathetic block.

Mourot et al. (2004) [13] have compared the traditional time and frequency-domain analysis assessment of HRV with the nonlinear Poincaré plot analysis of RR in healthy sedentary subjects, overtrained subjects, and trained subjects. This methodology was proposed to identify fatigue and/or to prevent an overtraining syndrome where a narrowing pattern was observed in the shape of the Poincaré plots. This distinctive shape formed by enhanced sympathetic and withdrawn parasympathetic activity was also detected in participants after supine and standing training, most likely due to increased parasympathetic activity.

Naranjo-Orellana et al. (2015) [15] proposed the Poincaré plot to evaluate the S/PS ratio for the follow-up of the individual assimilation of weekly workloads, using weekly recording of the HRV over a period of 10 min, and they discussed the validity of this method in monitoring the training status.

The Poincaré nonlinear method has many applications in the detection of the metabolic thresholds, as described in Section 4.

### 3.2. Detrended Fluctuation Analysis (DFA)

DFA is a useful algorithm that has proven useful in measuring the long-term autocorrelation of non-stationary time series. It can quantitatively characterize the complexity of the series using fractal theory [16,35,46].

In general, DFA measures the mean square deviation of a signal from its local trend line, on a variety of scales. The steps for the procedure are as follows:For a time series *RR(i),* the integrated time series *y(k)* can be expressed as:
(1)$$y\left(k\right)=\overset{k}{\underset{i=1}{\sum }}\left[RR\left(i\right)-\bar{RR}\right)]$$
where *RR(i)* corresponds to the *i*-th point of the RR time series and the $\bar{RR}$ is the average of all the points.

2.The time interval is divided in windows with equal length n, to quantify the vertical characteristic scale of *y(k);*3.In each window n, a least-squares line is fit to the data (representing the trend in that window). The y-coordinate of the straight-line segments is denoted by *y_n_(k).* The integrated time series is detrended, *y(k),* by subtracting the local trend, *y_n_(k),* in each window (see Figure 1b in [35]).4.The root mean square fluctuation of the integrated and detrended time series is estimated by the formula:


(2)
$$F(n)=\sqrt{\frac{1}{N}\sum_{k=1}^{N}\left[y\left(k\right)-y_{n}\left(k\right)\right){]}^{2}}$$


The procedure is repeated by dividing the time series into time-windows with different lengths. After that, the log-log scale of the F(n) versus the window size n is depicted. Representing the function F(n) in a log-log diagram, two parameters are defined: α1, as the slope of the regression line relating log(F(n)) to log (n) with n usually comprised in the interval from 4 to 16; α2, as the slope of the regression line relating log(F(n)) to log(n) with n in the interval from 16 to 64. The fractal dynamics of the time series are represented by the existence of a linear relation between F(n) and the number of parts. The value of α provides information about the series self-correlations:if 0 < α < 0.5, then the process exhibits anti-correlations;if α = 0.5, then the process is a random process, such as white noise;if 0.5 < α <  1, then the process exhibits positive correlations;if α= 1, a long-range correlation in the time-series occurs, corresponding to the typical 1/f noise, where f is the frequency. The 1/f slope of the log(power) vs. log(frequency) plot was obtained from a linear regression;if α > 1, then the process is non-stationary;if α  =  1.5, the process represents a random walk such as Brownian noise.

The slope values, α1 (or DFA-α1) for long correlation (in green) and α2 (or DFA-α2) for short correlation (in blue), are shown in Figure 4.

The potentiality of DFA on HR analysis was confirmed by Goya-Esteban (2012) [19], who suggested how this nonlinear method could assess relevant information about physiological recovery, which is the time period between the end of exercise and the subsequent return to resting state in an all-out exercise test in eight male amateur triathletes, during and after high intensity exercise.

Chen et al. (2015) [16] introduced an innovative time-varying parameter called the cardiac stress index (CSI), which is similar to the stress score introduced in the Poincaré plot explanation. The CSI is defined as the ratio of the number of events with α < 1 to the total number of events, and it may be used to represent the progressive status of cardiac stress continuously and quantitatively.

The DFA method was shown to have relevant applications in the detection of metabolic thresholds [47,48], and the information is presented in Section 4.

### 3.3. Entropy

Entropy quantifies the degree of order in a signal (regular, predictable, and limited complexity systems produce signals with low entropy). Two main entropy measurements can be used to assess the complexity of biological signals, such as heart rate, respiration, or EEG signals [49]. These are approximate entropy (ApEn) [50,51] and sample entropy (SampEn) [38]. Sample entropy (SampEn) is a modification of the approximate entropy (ApEn), introduced for the first time by Pincus (1991) [36], and used for assessing the complexity of physiological time series signals. In the following, the definitions of these measurements are provided.

#### 3.3.1. Approximate Entropy (ApEn)

A brief description of the algorithm for computing approximate entropy is reported. Starting from a time series of RR intervals of length N, the maximum absolute difference between their respective scalar components is calculated as the distance D[X_i_,X_j_] between vectors X_i_ and X_j_, where X_i_ = [RR_i_, RR_i+1_… RR_i+m__−1_] is the i-th vector of length m constructed from the N RR intervals RR_1_, RR_2_, …, RR_N_. For each vector X_i_, B_i_ is the relative number of vectors X_j_, (with j = 1,2, N − m + 1) for which D[X_i_,X_j_] ≤ r, where r is a tolerance value (or “radius” in the following); then, $C_{i}^{m}\left(r\right)$ is computed.

The count Bj(r) is the number of vectors X_m_(j) within r of X_m+1_(i), and $C_{i}^{m}\left(r\right)$ is defined as the ratio:(3)$$C_{i}^{m}\left(r\right)=\frac{Bj\left(r\right)}{N-m+1}$$

By taking natural logarithm of each $C_{i}^{m}\left(r\right)$, the index Φ^m^(r) is reckoned:(4)$$\Phi^{m}\left(r\right)=\frac{1}{N-m+1}\overset{N-m+1}{\underset{i=1}{\sum }}ln\;C_{i}^{m}\left(r\right)\;$$

Finally, ApEn expresses the logarithmic probability that the signal repeats within the tolerance of r for both m and m+1 points.
(5)$$ApEn\left(m,r,N\right)=\Phi^{m}\left(r\right)-\Phi^{m+1}\left(r\right)$$

Higher ApEn values suggest a more complex time series structure [37,49,50,51,52].

#### 3.3.2. Sample Entropy (SampEn)

The SampEn is a modification of the approximate entropy (ApEn) since (i) the self-matches are excluded in the calculation of Bj(r), and (ii) to ensure that the number of RR intervals used for $Bj{\left(r\right)}^{m}$ is the same as the number of RR intervals available for $Bj{\left(r\right)}^{m+1}$, for a fixed m, only the first N–m RR intervals in Xj of length m are used for calculating $j{\left(r\right)}^{m+1}$:(6)$$C_{i}^{m}\left(r\right)=\frac{Bj{\left(r\right)}^{m}}{N-m}$$
(7)$$C_{i}^{m+1}\left(r\right)=\frac{Bj{\left(r\right)}^{m+1}}{N-m+1}$$

$C_{i}^{m}\left(r\right)$ is an estimate for the probability that two different sequences of RR intervals will match for m points, whereas $C_{i}^{m+1}\left(r\right)$ estimates for the probability that two different sequences will match for m+1 points.
(8)$$SampEn^{m}\left(r\right)=-ln\;\frac{C_{i}^{m+1}\left(r\right)}{C_{i}^{m}\left(r\right)}\;$$

The sample entropy is essentially the negative logarithm of the ratio of the probability that any two different sequences in a time series match for m points to the probability that any two different sequences in the time series match for m + 1 points [47]. Since $C_{i}^{m+1}\left(r\right)$ will always be smaller or equal to $C_{i}^{m}\left(r\right)$, SampEn will always be either zero or positive. A smaller value of SampEn in the time series indicates more self-similarity in the time series or less noise.

In a descriptive longitudinal cohort studied by Orellana et al. (2015) [35], the behavior of SampEn during a work accumulation overtime (i.e., throughout the season) was observed in high level athletes (soccer players). The group values of SampEn were always above 1, and with a clear tendency for stability throughout the season.

For our knowledge, there are no reports in the literature of any entropy applications in the detection of the metabolic thresholds.

### 3.4. Recurrence Quantification Analysis (RQA)

Statistical methods have been widely used to assess signals generated by adaptive biological systems, by using different approaches to describe physiological changes, including ANS activity, in terms of phase transitions [30,53].

RQA is a statistical, graphical and analytical tool [30,32] for studying nonlinear dynamical systems that has been effectively applied in a variety of fields, including physiology [10,54,55,56], earthquakes geoscience [57,58] and finance [59]. RQA was shown to be a reliable technique in analyzing the chaotic and stochastic dynamics of the physiological signals [60].

The procedure to obtain the RQA measures is briefly described here below:

Starting from a time series of RR intervals, the vectors X_i_= (RR_i_, RR_i+τ_,…, RR_i+(m−1)_·_τ_), with i = 1,…, K, with K = [N − (m − 1)×τ)], are defined, where m is the embedding dimension and τ is the embedding lag. RR_i_ represents the RR value corresponding to the i-th time position. Then, the series was arranged in successive columns (the columns number is defined by the embedding dimension m), each one obtained by applying a delay in time (lag parameter) to the original sequence, thus creating an embedding matrix, as in Equation (9) (for τ =2).
(9)$$\left[\begin{array}{c} \begin{array}{ccc} RR_{1} & RR_{3} & \ldots \;\\ RR_{2} & RR_{4} & \ldots \\ \ldots & \ldots & \ldots \\ RR_{N}{}_{} & & \end{array} \end{array}\right]=\left[\begin{array}{c} \begin{array}{cccc} e_{1,1} & e_{1,2} & \ldots \; & e_{1,m}\\ e_{2,1} & e_{2,2} & \ldots & e_{2,m}\\ \ldots & \ldots & \ldots & e_{3,m}\\ e_{N-2m+2,1}{}_{} & e_{N-2m+2,2} & \ldots & e_{N-2m+2,m} \end{array} \end{array}\right]$$

The embedding matrix is generated by representing the series in column format (RRi in left side) and copying it into m consecutive columns, each one displaced by a number τ of points (e_1,1_ = RR_1_, e_1,2_ = RR_3_…, on the right side).

If the distance between the j-th and i-th rows of the embedding matrix is smaller than a predetermined value (known as the radius or cut-off [30,61,62]), the value 1 is associated to the element (i,j) of the distance matrix. If instead of associating a binary value, the value of the distance is associated, the distance matrix is called an unthresholded recurrence plot (Figure 5). The horizontal and vertical axes of the recurrence plot are the i and j indexes.

The number and placement of dots (called recurrence points) on the graph are used to calculate the RQA measures (Figure 5, on the left side). The percentage of determinism (DET) and laminarity (LAM) are two main parameters that quantify the specific characteristics of the time series.

The percent of determinism (DET) is the percentage of recurrence points which form diagonal lines (l), and it indicates the degree of deterministic structure of the signal. It is usually an indicator of regime changes and phase transitions [63]:(10)$$DET=\frac{\sum_{l=lmin}^{K}l\;P\left(l\right)}{\sum_{l=1}^{K}l\;P\left(l\right)}$$
where *P*(*l*) is the histogram of diagonal lines *l* in the RP and *lmin* is the minimal considered length of a diagonal line; laminarity (LAM) is the percentage of recurrence points forming vertical line segments in the recurrence plot and indicates the fraction of laminar phases (intermittency) in the tested system:(11)$$LAM=\frac{\sum_{l=vmin}^{K}v\;P\left(v\right)}{\sum_{l=1}^{K}v\;P\left(v\right)}$$
where *P*(*v*) is the histogram of vertical lines *v* in the RP, and *vmin* is the minimal considered length of a vertical line. More details can be found in [62,63,64] and [9,30,31].

The investigation of the evolution of the RQA measures over time was obtained by using windows that are shifted along the RP’s major diagonal (the RQ epoch-by-epoch method is called RQE). This allows the detection of transitions in dynamical systems in different application fields [45,57,58,59,60,61,62]. DET and LAM reckoned by RQE are used for metabolic threshold detection in [65] and in [66] (see Section 4.2.2). From a theoretical point of view, the ability of RQA to predict abrupt changes is in line with the fact that RQA is based upon changes in the correlation structure of the observed phenomenon that is known to precede the actual event. In the Figure 5, three different patterns are recognizable.

Gorban et al. (2010) [67] studied the behavior of systems approaching a critical transition in a wide range of conditions ranging from physiology to ecology and finance. In all these cases, they observed that a drastic increase of the internal correlation, together with an increase in variance, appeared before the crisis symptoms. The RQA is totally independent from stationarity constraints; it does not need long streams of data and it is focused on the quantification of correlation structure of the studied dynamics. For these reasons, it is particularly suited to locate transition behavior in signals.

RQA is a statistical approach that converges to the same results of theoretical ones [68,69,70] without the need of any a priori hypothesis [71]. In the last few years, new papers have used the same or similar language to describe noise-induced critical transitions in natural, social and technological systems, which can lead to detrimental system breakdowns [70]. Statistical physics concepts and methods are commonly used in the complex natural system, such as the Earth, as recently discussed in Fan et al. (2021) [72]; Pitsik et al. (2020) [54] reported that the RQA method was developed to numerically investigate EEG by using various complexity measurements. Recently, Hasselman (2022) [71] explained that the common purpose of RQA is to characterize the dynamics presented during the observation time, rather than to infer a true model, process or estimate a population model parameter.

One of the first applications of RQA on RR time series suggested that the cardiac age may be revealed from healthy individual tachograms (RR time series) [31]. This approach has successfully predicted the age of a hundred participants based on a gradual increase in the deterministic character of the heartbeat. This allowed for the development of a simple and comprehensive model of the aging effect on cardiac dynamics: HR dynamics become more predictable (constrained) on a beat-by-beat basis as age proceeds [31]. Hoshi et al. investigated the HR complexity of healthy and young individuals after recovery from maximal exercise [29]. The regularity of the HR rhythm was shown to increase after maximal exercise, while the signal’s complexity decreased throughout the recovery phase. After 80 min, these conditions progressively reverted to rest levels and reached recovery. Singh [17] found that the RQA complexity index can significantly discriminate between age groups even from small data sets. These results corroborate the findings of previous studies [9,10,31].

The RQA enabled a robust method to identify laminar states and their transitions to a regular as well as to other chaotic regimes in complex systems [65]. Indeed, these transitions can be only qualitatively distinguished by spectral analysis.

RQA analysis was recently included in different commercial or open-source software such as Kubios [73], RHRV [74] and RR-APET [75]. Physionet, a web portal containing an extensive database of physiologic signals as well as software tools [76], has already announced the possibility to perform different HRV analysis. Additional software tools for HR analysis are reported in a paper by Rodriguez-Linares et al. (2014) [77].

As it is shown in Section 4, some differences occur when the test is performed in a laboratory or in an open field setup, where the experimental constraints are controlled with more difficulty. For example, a lower accuracy in the data collection was observed to have a considerable impact on the derived results.

In summary, RQA measures can successfully describe variations in ANS activity and localize dynamical transitions [65,66]. The encouraging applications on metabolic thresholds detection will be discussed in the next section.

## 4. HR Time Series Analysis for Detection of Metabolic Thresholds during Exercise

In this section we describe the application of the HR time series analysis methods for the detection of the metabolic thresholds, and their applications. HRV is the most used measure. A comprehensive and clear explanation of investigations concerning the interrelation between HRV and ANS is reported by Dong (2016) [76], who examined how the application of HRV on physical exercise may play a significant role in sports physiology. HRV can be considered a performance indicator, such as the VO_2max_, the power, the rating of perceived exertion (RPE) and the blood lactate concentration (BLa), which classify individuals and set their training zones [77]. Furthermore, in HRV studies related to metabolic thresholds and exercise physiology on athletes, it is needed to describe accurately the analysis methods, the population samples, the training protocols and the intensity and duration schedules [78,79]. Buchheit (2007) [80] stated that most of the contradictory findings are due to methodological flaws and/or data misinterpretation, rather than limits of HR measurements to effectively inform on training status.

The 13 papers selected by using Covidence Online Software (see Additional info of Section A.4) regarding the detection of metabolic thresholds through HRV analysis are presented in chronological order in Table 1. In the following sections, we will distinguish feature- based methods (4.1) and non-linear methods (4.2) for the detection of metabolic thresholds. The original name chosen by authors for the parameters are generally conserved in the text.

### 4.1. Feature-Based Methods for the Detection of Metabolic Threshold

Buchheit et al. (2007) [81] used spectral analysis and compared the accuracy of the HR deflection point (HR_DP_) and the second HR variability threshold (HRV_Th2_) to predict anaerobic threshold (AnT) in 72 boys during an incremental test to exhaustion. The test was conducted on a treadmill with the grade set at 1%; the initial velocity was 6.0 km/h and was linearly increased by 1 km/h every minute. A chest belt was used to continuously record beat-to-beat HR during exercise. The deflection point was determined from slope trends of successive linear regressions. HRV_Th2_ was visually determined from the high frequency’s peak HF_p_ and power density trends (*ln* fH_Fm_./HF_p_); the VSFT method was carried out with a 64-s moving window and a time shift of 3 s. No differences were found in VO_2_ or HR, and a significant relationship between these variables at AnT was found (HRV_Th2_ versus HR_DP_ r = 0.88 and r = 0.85 for VO_2_ and HR, respectively, *p* < 0.001)

Taking into account the original RR intervals, the standard deviation of the RR interval (SDNN) was found to be closely related to the AerT, but no verification studies have been conducted [82]; in this context, Karapetian’s study (2008) [82] has demonstrated a marked RR interval deflection point in the stage of exercise when the lactate threshold took place (determined via the BLa blood levels). Twenty-four adults performed a graded maximal test on a cyclergometer; the exercise intensity started at 25 W, and every 3 min the intensity increased at 25 W increments. The RR intervals of the last 2 min of each stage were used for the analysis of HRV. Time-domain indices such as the standard deviation (SDNN), the mean successive difference (MSD) and the mean absolute difference between consecutive RR intervals have been shown to strongly correlate with the vagal tone during the exercise (r = 0.87 and 0.92, respectively; *p* < 0.001). The visual detection of the HRV threshold (HRVT) consists of searching for the HRV deflection point using the MSD method, which was calculated for each exercise stage. To determine the HRVT, the MSD and SDNN of HR intervals for each stage of exercise were graphically plotted against the work rate. The results for the determination of HRVT during incremental exercise testing, using RR interval data, showed similarities in VO_2_ (L/min) values between LT detection and HRVT detected by the HRV deflection point. (HRVT1 vs. VT1 r = 0.89 for VO_2_, and HRVT2 vs. LT2 r = 0.82 for VO_2_, *p* < 0.001)

In 15 healthy young soccer players, Stergiopoulos et al. (2021) [89] explored the feasibility of estimating the second ventilatory threshold VT2 (AnT) through different HR variability indices during an incremental laboratory test and a multistage shuttle run test (MSRT) until exhaustion. ANOVA, *p* = 0.45, found no statistically significant differences between VT2 and the threshold established using both the high frequency product (HFp) and ECG-derived respiration (EDR) (see the Appendix A session, for EDR). The incremental test was conducted on a treadmill with 1-min phases and a starting speed of 8.0 km/h. Up until exhaustion, the speed increment between each stage was 0.5 km/h.

The patients ran back and forth on an indoor wooden surface between two 20-m markers as part of a one-minute procedure for the MSRT. This test was divided into 1-min sections with a starting speed of 8.0 km/h and 0.5 km/h increments every minute. The second ventilatory threshold was determined to be the point of the first abrupt increase in HFp after it had reached a minimum (followed by variable values) (HRVT2).

Quinart et al. (2014) [83] used both time-domain and frequency-domain analyses to estimate AerT and AnT from HRV in their investigation of metabolic threshold detection; the root mean square of successive differences (RMSSD) was performed using the RR intervals of the last 60 s of each stage in an incremental exercise test on a cycle ergometer. The RMSSD values move on a decreasing curve until they reach a point of stabilization. A higher correlation with the gold standard (gas exchanged) was found in 20 obese adolescents.

First and second thresholds (HRVTS1 and HRVTS2) were calculated using time-varying (average value every 3 s) functions that, respectively, corresponded to the first non-linear increase after reaching a minimum and to the last abrupt increase of the function ln(fHFm *HFp). To remove artifacts, HF peaks were modelled using a third order equation (fHFm). For both thresholds, a high Pearson’s correlation was obtained (HRT1 vs. VT1 r = 0.91, 95CI % (0.84–0.95) for HR and HRT2 vs. VT2 r = 0.91 95% CI (0.83–0.95) for HR (*p* < 0.001)).

One of the rare studies based on the “field” tests was the study performed on a skiing track described by Cassirame et al., 2015 [84], where the skiing track was 950 m long with an ascent of 168 m and an average gradient of 9.1°. The skiers began at 3 km/h and the speed was increased by 0.5 km/h every minute until exhaustion. Beat-by-beat RR intervals were recorded by the Polar^®^ T61 sensors, respectively. The metabolic threshold was estimated by fHF*HF_p_ and was compared with the VT1 and VT2 averaged values (for every 10 s), estimated by the “respiratory equivalent” method [2]. HF_p_ and the product of fHF*HF_p_ were plotted against time; the first threshold estimate (VT1_fH_) was determined as the first abrupt increase in fHF*HF_p_ after it had reached a minimum, while the second threshold (VT2_fH_) corresponded to the final abrupt increase [44]. Cassirame et al., (2015) [84] found high correlations between HRT2 vs. VT2 for HR (r = 0.91) and speed (r = 0.92), with small limits of agreement LoA (3.6 bpm for HR). The gas exchanged (being the gold standard) were valued by a portable measurement system (Cosmed K4b2, Rome, Italy) [92].

Vasconcellos et al. (2015) [85] estimated the anaerobic threshold HRVT1 by analyzing the RMSSD values of RR intervals within each 60 s stage, through CPET in 35 adolescent subjects (in whom 15 were obese). Their methodology followed the same approach used by Quinart (2014) [83]. The HRVT1 was defined as a HRV deflection point, below which there was no-further reduction in the RMSSD values (suggesting a parasympathetic withdrawal). The initial load in CPET was set at 25 W with an increment of 10 W/min until exhaustion. Correlations between HRVT1 and the gold standard (gas exchanged) VT1 for VO_2_ and VO_2_ reserve ranged from 0.89 to 0.95 (*p* < 0.001), and test–retest reliability ranged from 0.59 to 0.82 (*p* < 0.006).

Due to its strong connection with the inhibition of the vagal and activation of the sympathetic ANS, HRV can even predict the second metabolic threshold (AnT).

Ribeiro et al. (2018) [86] identified the second threshold in 15 young soccer players, at the point where the RR time series shifted during the incremental test CPET. The data were independently identified by two researchers. A ramp protocol was applied with an initial velocity of 7 km/h and increments of 1 km/h every minute until exhaustion. Even in this work, the gas exchange method was used as the gold standard technique. A Polar^®^ F11 chest belt was used to record the HR simultaneously to physical exercise. The RR intervals were determined by applying the equation: RR = 60000/HR. The HRV second threshold (here named RRiT2 = HRVT2) was independently identified, by two researchers, at the point where there was a shift in the RR curve. Ribeiro et al. (2018) [86] found positive correlations for all parameters predicted at the AnT (for time r = 0.84; for HR r = 0.97 and for VO_2_ r = 0.97), and they observed that those values were lower than those determined by the gold standard.

In summary, these studies applied linear methods and were mainly conducted in laboratory conditions; only a few studies were based on the “field” tests, and they used the gas exchange method as the gold standard. All studies used a chest belt to collect heart rate time series. Since different test protocols and different statistical methods were used, comparison is difficult; however, further useful information about research questions can be extracted from the obtained limitations.

### 4.2. Nonlinear Methods for the Detection of Metabolic Threshold

#### 4.2.1. Poincaré Plots and DFA for the Detection of Metabolic Threshold

By using Poincaré Plots, all measures of HR vagal modulation progressively decreased until the ventilatory threshold level was reached when sympathetic activation was reflected as changes in the SD1/SD2 ratio [87,88]. Nascimento et al. (2019) [87] have applied the Dmax method to estimate the lactate threshold (AnT), during a maximal incremental running test. They proposed a new approach based on SD1 and SD2 by using the Dmax method firstly introduced by Cheng et al. (1992) [93]. The Dmax method calculates the point that yields the maximal distance from a curve representing the ventilatory and the metabolic variables as a function of oxygen uptake (VO_2_) to the black line formed by the two end points (orange) of the curve (Figure 6).

The AerT and AnT were estimated during a maximal incremental running test in 19 male runners [87]. All participants used a cardio belt for beat-to-beat heart rate measurements. The treadmill test started at 8 km/h, which was increased by 1 km/h every 3 min, until exhaustion. The HR was recorded continuously throughout the assessment. RR interval data were estimated from the last 60 s of each stage of exercise. The gold standard used blood lactate methods. The identification of LT1 by the Dmax method on HRV data was demonstrated, and no statistically significant differences were observed between test and re-test (*p* = 0.55). For additional details, see Table 1.

Novelli et al. (2019) [88] evaluated the reproducibility of the HRVT by different HRV indexes (RMSSD and SD1) in 68 untrained young individuals, tested twice (non-consecutive days with a 72-h interval, maximum incremental cycle ergometer test): no significant differences were found (*p* < 0.05) between the test and retest for any of the variables evaluated. The test started with a workload of 15 W which was increased by 15 W per minute, keeping 60 RPM until volitional exhaustion. The RR intervals were recorded continuously (by Polar^®^ RS800CX); the metabolic thresholds were determined with these two criteria: the first one was identified by the physical effort intensity where the values of the SD1 and RMSSD index were < 3 ms and were < 1 ms, respectively.

Another example of the Dmax method is the work of Afroundeh et al. (2021) [91]. They assessed the agreement between HR deflection point (HRDP) variables with the gold standard (maximal lactate steady state, MLSS, at second threshold), in young males with different body mass indices (classified in four different groups). The HRDP was determined using a modified Dmax method. The 103 young male subjects were tested via a standard running incremental protocol, with an individualized speed increment from 0.3 to 1.0 km/h. Beat-by-beat HR (sampling rate of 1 Hz) was measured continuously during the exercise tests by Polar^®^ T31 sensor. If the individual was more active and/or of normal weight, the speed increment was set to a higher limit in order to enable him to stay within the target completion time of 12–16 min. If the subject was obese, the speed increment was of 0.3 km/h. A good agreement was observed between HRDP and MLSS for HR for all participants (*p* < 0.001).

In Rogers et al. (2021) [90], the short-term scaling exponent α1 determined by DFA has been shown to steadily change with increasing exercise intensity. The study included data from 17 male adults, via an incremental treadmill test. The speed and inclination started from 2.7 km/h at a 10% grade, increased by 1.3 km/h and 2% grade every 3 min, until exhaustion. It was found that an α1 value of 0.75 was directly associated with the first ventilatory threshold and an α1 value of 0.50 with the second one. This is a rare situation where a fixed value was suggested as a means to identify the thresholds. Based on the Bland–Altman analysis and linear regression, a strong agreement between VT and HRVT measured by HR (for the first threshold r = 0.96, *p* < 0.001; for the second threshold r = 0.78, *p* < 0.001) was found.

#### 4.2.2. RQA for the Detection of Metabolic Thresholds

Recently, the RQA has proposed a simple landscape where a description of the cardio-respiratory physiological coupling is available under an increasing physical effort in terms of complexity and dynamical transitions. The change in the RP pattern was evident in Figure 5 (as in Figure 7 of [65] and Figure 1 of [66]). The RQA-based approach was validated, using both determinism (DET) and laminarity (LAM), to identify the ventilatory thresholds [65,66]: the assessment of the AerT threshold (red line in Figure 5) occurs upon the identification of a DET minimum (DET_min_) confirmed by the corresponding minimum of LAM (LAM_min_). The AnT threshold (green line Figure 5) is detected when the DET value reaches a saturation plateau [66]. In the epoch-by-epoch analysis (the window size was w = 100 data points), the delay in the embedding procedure (lag) was set to 1, the embedding dimension was 7, the cut-off distance (radius) was 50 and the diagonal and vertical line minimum was 4 (l min = v min). The RQA was used to detect the first metabolic threshold [65] from HRV during a graded incremental physical exercise. The epoch-by-epoch analysis permitted the detection of a chaos-order transition of the time series dynamic, which corresponded to the aerobic threshold. Additional results were obtained in subjects with different levels of fitness, and it was demonstrated that even the second threshold is detectable by RQA from HR [66]. The ordinary least products (OLP) regression analysis showed that, at both thresholds, the RQA and the gold standard methods (gas exchanged) had very strong correlations (r > 0.8) across all variables (HR, VO_2_ and workload).

In summary, it can be observed that the different studies reported in this section were characterized by a higher homogeneity in the protocol of physical exercises tests and in the starting hypotheses. In the following paragraph, a detailed explanation about their main limitations will be presented, showing the use of non-linear methods in new applications offered by the synergic complement between nonlinear methods and “big data” approaches. The first results (real time monitoring) can be implemented in the near future in wearable devices.

## 5. Known Issues in Using Nonlinear Methods

This section discusses some relevant issues in the evaluation of the metabolic thresholds from HR data by nonlinear analyses (Figure 7):(i)The intrinsic individual variability of the *subject* and the dependence by his/her fitness level and the body mass status;(ii)The accumulated sampling error and the noise levels of the *devices* used to record the HR data (sampling rate, motion, for both ECG or HR beat-by-beat monitor);(iii)The intrinsic features of the recorded *signal*, as non-stationarity or the trend;(iv)The dependence on the parametric values for the chosen analysis *method* (i.e., input values chosen for the algorithms, type of detrending, etc.).

A significant limitation in HR time series analysis is the high individual variability. The individual variability of the cardiac signal is a common problem of all physiological systems. In RQA-based methods, individual amplitude differences do not disturb the time series features, given each single RQA is auto-scaled on its proper standard deviation. The possible sources of uncertainty in determining the thresholds may originate from the exercise protocol, used to produce a specific workload, and that is also related to the subject’s fitness level. As it is explained in the introduction, the stationary hypothesis is an intrinsic limitation in most of the techniques of analysis used, except for DFA and RQA.

There are various reasons why the different approaches have been found to be inefficient in determining metabolic thresholds, even if the relevant aspects generally concern the quality of the acquired signals. The epoch-by-epoch analysis with sliding windows was used to observe the change of complexity in time [66], and it was useful even in reducing the noise where, in the incremental exercises, the workload changed every 60 s [65]. The epoch-by-epoch analysis was carried out on superimposed windows of 100 points, shifted by one point (about 2 s) [66].

Another issue in the HRV analysis during incremental exercises is the presence of a trend that can cover subtle but fundamental structures of the signal. In previous studies [65,66], the authors have analyzed HRV recorded during incremental physical exercise by a chest strap (Polar^®^ sensor, connected with the device Quark RMR-CPET Cosmed™, Rome, Italy), and a strong linear correlation with the linear increase of the workload was found. To remove this trend, a linear fit to detrend HR data was chosen; since it is a robust approach avoiding data overfitting analysis, initial rest and recovery (the test is sub-maximal, thus the VO_2max_ values were excluded) were not considered; the equation of the trend line was found and, from the original time series, the theoretical one was subtracted point by point: in this way, the final new series corresponded to a detrended signal [65,66].

Another interesting issue was raised by Henriques et al. (2020) [34]. They discussed the fact that most algorithms are parametric; therefore, the choice of the input parameters can influence the results. The procedure to mitigate this limitation is to run the algorithms with different input parameter values and choose those maximizing the differences in the observed output measurements.

A variety of computational approaches have been developed to assist specialists in providing an objective and time-efficient determination of ventilatory thresholds; however, the currently existing algorithms require pre-processing actions and are extremely sensitive to the signal-to-noise ratio.

Recently, an alternative approach to detect thresholds, but not including the HR time series, was proposed by Zignoli et al. 2020 [94]. CPET data interpretation has been improved using artificial intelligence (AI) technologies such as machine learning. This approach could reduce evaluation errors and the variability of interpretations across different experts and centers. It is built on a network of internationally recognized professionals in the field of ventilatory thresholds by CPET. Previously collected datasets are utilized to train a convolutional neural network for the interpretation of new data files. Since it would be impossible to evaluate the impact of an inaccurate evaluation on the whole algorithm performance, this approach can be nowadays used only in parallel with human visual inspection.

## 6. Conclusions

The gold standards methodologies adopted to detect metabolic thresholds [1,2] suffer from major application limitations that severely restrict their dissemination to large segments of the population.

To overcome some of these limitations, complementary and alternative procedures based on the analysis of HR time series have been suggested. The rationale behind these approaches lies in the observation that metabolic adaptation due to external stimuli is mediated through a complex regulation of cardiovascular and cardiorespiratory systems.

Cardiovascular and cardiorespiratory systems very likely interact with each other in a nonlinear way that can be detected by monitoring HR during an incremental physical exercise [9,32,62,95]. Indeed, under the hypothesis that fatigue is a conscious interpretation of the homoeostatic, centrally controlling mechanisms [66], it can be considered an order parameter in the chaos–order transition [65].

Time-domain descriptors of HR dynamics, such as RMSSD, HF or SD1 of the Poincaré plot, can be determined and have been proved to provide relevant insight into cardiac parasympathetic regulation [13,32,39,52]. Accordingly, the cardiac sympathetic activity that generates highly nonstationary rapid responses cannot be described by standard HR time series data analysis methods [52].

In addition, as several nonlinear methods (DFA and RQA) do not request stationarity, long data series and pre-processing data manipulation have been largely adopted to monitor physical exercise and to detect metabolic thresholds.

In a sports-context, the short- and long-term DFA components well describe the complexity, although short-term components have been found to be more reliable than the long-term ones [47,48,52,96]. DFA α1 has been also used to evaluate aerobic and anaerobic threshold [87,88,89,90]. This method is reliable in determining thresholds and independent on test protocols/workout, but requires a strong preprocessing of HRV data to avoid artefacts.

Moreover, RQA-based methods have been adopted to explore HR response during incremental exercise for the localization of metabolic transitions [65,66]. These approaches, while requiring minimal data preprocessing and allowing an immediate visual localization of the metabolic transitions, were found to be robust and reliable. The threshold points can also be quantitatively determined as they correspond to an abrupt change in determinism and laminarity, detected by DET_min_ and LAM_min_, respectively, in the epoch-by-epoch RQA procedure (see the Section A.3). At both thresholds, OLP regression analysis had very strong correlations between RQA and the gold standard across HR, VO_2_ and workload.

In conclusion, the nonlinear analysis of HR time series can be considered a valid alternative metric to assess metabolic thresholds providing easy-to-read and -use parameters to medical and paramedical staff. Furthermore, since HR data can be easily acquired by many low-cost devices, including smartwatches, the large widespread usage of nonlinear analysis of HR time series for a personalized evaluation of fitness level and planning/monitoring of training or rehabilitation procedures can be reasonably expected.

## Figures and Tables

**Figure 1 ijerph-19-12719-f001:**
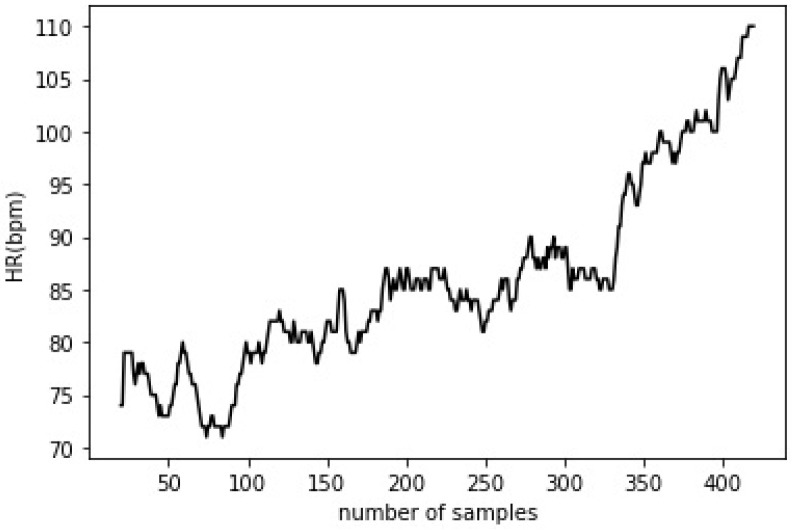
Heart rate variation (in beats per minute—bpm) of a healthy female subject during an incremental exercise (for more details see Section A.4).

**Figure 2 ijerph-19-12719-f002:**
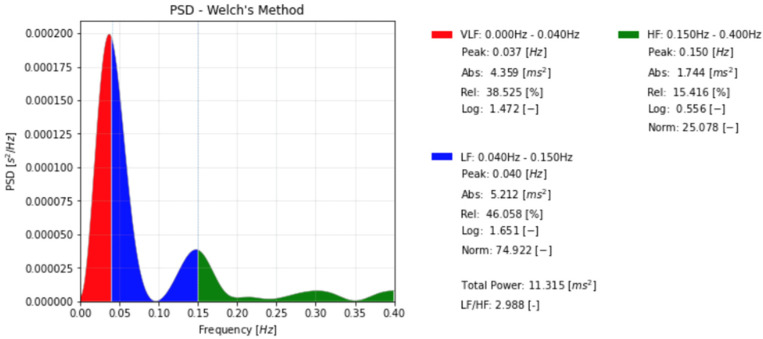
Spectral analysis of heart rate (HR) of a healthy female subject (the same as shown in Figure 1, for more details see Section A.4).

**Figure 3 ijerph-19-12719-f003:**
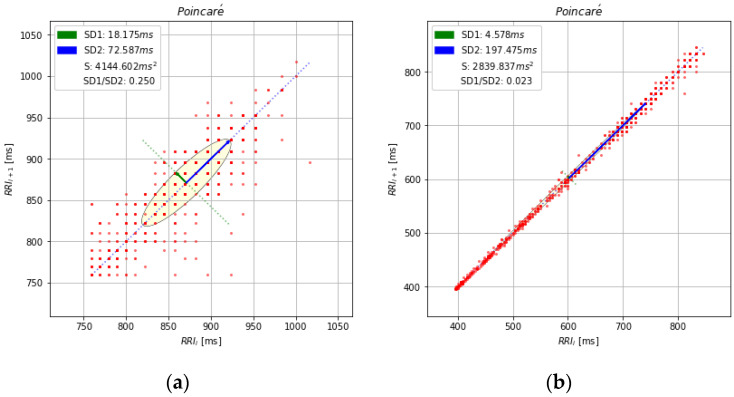
(**a**) Poincaré plot of RR time series from a healthy subject at rest; (**b**) Poincaré plot of the same time series as in Figure 1 (for more details see Section A.4).

**Figure 4 ijerph-19-12719-f004:**
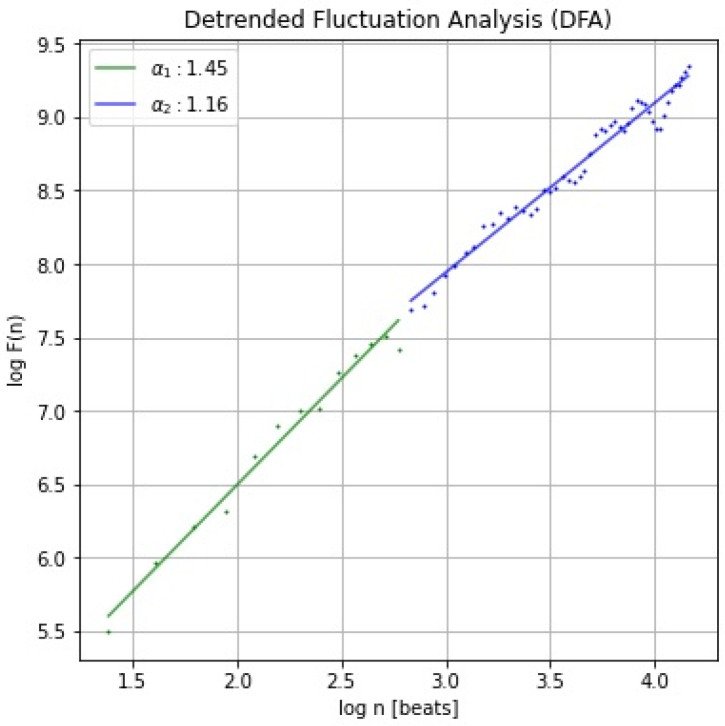
Detrended Fluctuation Analysis (DFA) of RR time series from a healthy subject (the same as shown in Figure 1, for more details see Section A.4).

**Figure 5 ijerph-19-12719-f005:**
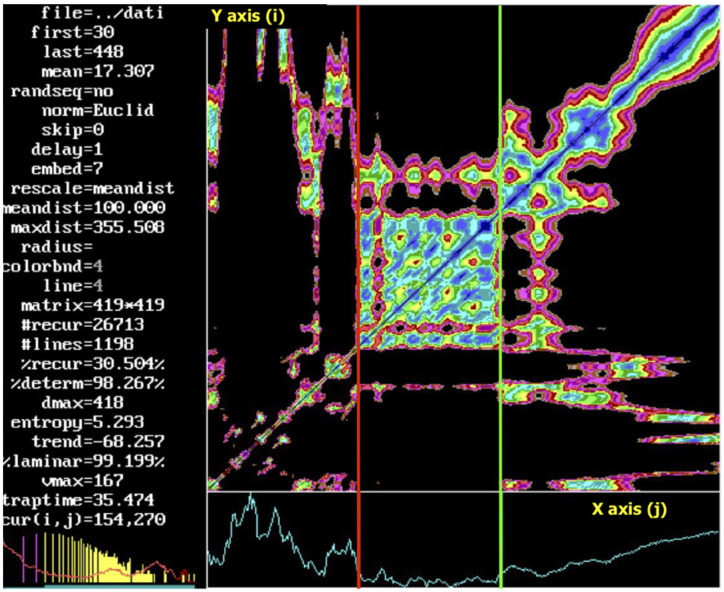
Unthresholded recurrence plot of the RR time series (ms) (light blue line) recorded as a breath-by-breath, from a cardiopulmonary exercise test (CPET) device (Cosmed, Rome, Italy). The red and green lines show the change of pattern at the first and second threshold, respectively. On the horizontal and vertical axes, the j-th and i-th indices are reported, respectively. (For more details see in Section A.3 and Section A.4).

**Figure 6 ijerph-19-12719-f006:**
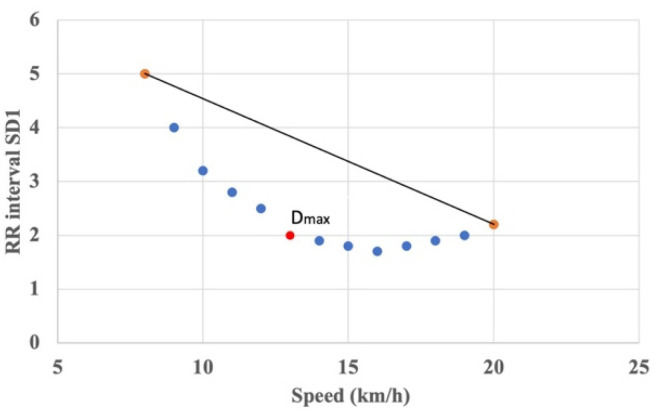
How the threshold (lowest red point) is determined by the Dmax method: it is estimated by the longest perpendicular distance between SD1 (predicted by a third order polynomial function over actual value) from the linear regression calculated with the first and last values of the curve. The speed (km/h) corresponds to the treadmill velocity.

**Figure 7 ijerph-19-12719-f007:**
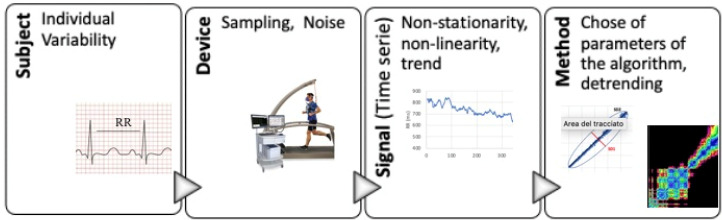
From the subject to the method: a presentation of the principal limitations in the HR data processing. In the insets: ECG, Experimental set: treadmill and facemask, Quark RMR-CPET Cosmed, Rome, Italy.

**Table 1 ijerph-19-12719-t001:** HR and metabolic thresholds.

Authors	Subjects	Physical Exercise	HR Detection	Methods	Statistical Validation	Gold Standard
Buchheit et al., 2007 [81]	72 Trained boys, runners	Treadmill	CPET Chest belt Polar T61; watch Polar 810s	Spectral analysis HFp and (*ln* fHFm./HFp)	HRVT2 vs. HRDP for VO_2_ r = 0.88 *; for HR r = 0.85 *	Gas exchange
Karapetian et al., 2008 [82]	24 Healthy adults	Cyclergometer	Watch and chest belt Polar Vantage XL	Time analysis SDNN, MSD	HRVT1 vs. VT1 for VO_2_ r = 0.89; HRVT 2 vs. LT2 for VO_2_ r = 0.82	Gas exchange Blood lactate
Quinart et al., 2013 [83]	20 Obese adolescents	Cyclergometer	CPET Chest belt and watch Polar 810s	Spectral analysis HFp and (*ln* fHFm./HFp) Time analysis RMSSD	HRT1 at VT1 for HR r = 0.91 ** 95% CI (0.84–0.95); HRT2 at VT2 for HR r = 0.91 ** 95% CI (0.83–0.95)	Gas exchange
Cassirame et al., 2015 [84]	9 Healthy adults ski-mountaineers	Alpine skiing track	Chest belt Polar T61; portable recorder FRWD B100	Time–frequency analysis fHF × HFp	HRT2 vs. VT2 for HR r = 0.91; for speed r = 0.92 for HR small LoA (3.6 bpm)	Gas exchange
Vasconcellos et al., 2015 [85]	35 Adolescents (15 obese)	Cyclergometer	CPET Chest belt and watch Polar TM RS800cx	Time analysis RMSSD	HRVT1 vs. VT1 for VO_2_ from r = 0.89 ** and test–retest reliability r = 0.59 *	Gas exchange
Ribeiro et al., 2018 [86]	13 Young soccer players	Treadmill	CPET Chest belt and watch Polar F11 HRM	Graphics analysis (shift of RR interval)	HRVT2 vs. VT2 for time r = 0.84 *; for HR r = 0.97 *; for VO_2_ r = 0.97 *	Gas exchange
Nascimento et al., 2019 [87]	19 Male runners	Maximal incremental running test (MIRT)	Chest belt and watch Polar 810s	Poincaré plot, DFA (Dmax)	No correlation for HR1 at LT1 for HR; HRT2 at LT2 for HR r = 0.71 **; HRT1 at LT1 for speed r = 0.46 *, 95% CI (0.9–1.9); HRT2 at LT2; for speed r = 0.48 * 95% CI (0.8–1.6)	Blood lactate
Novelli et al., 2019 [88]	68 Untrained subjects	Cyclergometer	Chest belt and watch Polar RS800CX RRinterval	Time analysis, Poincaré plot RMSSD and SD1	No significant difference (*p* < 0.05) between the test and retest for any of the variables. All variables at HRVT1 and the heart rate at HRVT2 showed CV ~10%.	Gas exchange
Zimatore et al., 2020 [65]	20 Obese adults	Treadmill	CPET Chest belt Polar RS 400	RQA	HRVT1 vs. VT1 for time r = 0.70 **; for speed r = 0.84 **; no statistically significant differences (*p* < 0.05) for time, speed, VO_2_, and MFO;	Gas exchange
Stergiopoulos et al., 2021 [89]	15 Healthy adults	Treadmill, multistage running test (MSRT)	ECG TEL100, MP 100A Biopac	Time and apectral analysis HFP	no statistically significant differences between the running speed at VT2 and EDRT (F (2,28) = 0.83, *p* = 0.45, η2 = 0.05)	Gas exchange
Zimatore et al., 2021 [66]	31 Healthy adolescents	Cyclergometer	CPET Chest belt Garmin ^ HRM-Dual™	RQA	HRVT1 vs. VT1 for HR r = 0.87 *; for workload r = 0.95; for VO_2_ r = 0.91 *; HRVT2 vs. VT2, for HR r = 0.89; for workload r = 0.97; for VO_2_ r = 0.97	Gas exchange
Rogers et al., 2021 [90]	17 Male adults	Treadmill	Chest belt Polar H7	DFA	HRVT1 vs. VT1 for HR r = 0.96 **; HRVT2 vs. VT2 for HR r = 0.78 **;	Gas exchange
Afroundeh et al., 2021 [91]	103 Young males	Treadmill	Chest belt Polar T31	DFA, modified Dmax	HRVT1 at LT1 for HR ICC = 0.88 **; for speed ICC = 0.40 **	Blood lactate

For sake of simplicity, the first and the second thresholds obtained by HR are named HRVT1 and HRVT2, respectively; those obtained by the gas exchange method are named VT1 and VT2, those obtained by blood lactate method are named LT1 and LT2; HRDP is HR obtained by the HR deflection point; LoA, limits of agreement; MFO, maximal fat oxidation (* *p* < 0.05; ** *p* < 0.001). * In the first column, the first author ref. is reported; ^ Olathe, KS, USA.

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
