# Peer review of "Detecting Metabolic Thresholds from Nonlinear Analysis of Heart Rate Time Series: A Review"

_ijerph, 2022, doi:10.3390/ijerph191912719_

Round 1

Reviewer 1 Report

I’d like to thank the Authors for the opportunity to review the article submitted to the International Journal of Environmental Research and Public Health.The study is very timely and focuses on a really important issue. I have no doubt that your manuscript will merit publication.

Congratulations of your hard work.
I only have a few editorial notes:
1. The parenthesis is missing from line 155. In addition a comma should be replaced with a dot. Therefore, it should be as following: ‘To overcome the lower HR modulation-stability Cottin et al. (2004) [18] proposed a’.
2. The dot is missing from the line 353. It should be: ‘Gorban et al. (2010) [67] studied the behavior of systems approaching a critical tran-‘.
3. The spaces is missing from line 484 for p values. It should  be: ‘0.91 95C I% (0.83-0.95) for HR (p < 0.001)’.
4. There is a non-uniform record, because once is 0.001 (for example on line 450) and once is .001 (for example on line 439). A uniform record is therefore necessary.

Best wishes.

Author Response

Thank you for your kind message concerning our manuscript entitled “Detecting metabolic thresholds from nonlinear analysis of heart rate time series”. 

In the following we list the reviewers’ comments (in black) and our answers (A: in blue in italics if a phrase of the text is reported)

REVIEWER 1:

I’d like to thank the Authors for the opportunity to review the article submitted to the International Journal of Environmental Research and Public Health.The study is very timely and focuses on a really important issue. I have no doubt that your manuscript will merit publication.

Congratulations of your hard work.

I only have a few editorial notes:

  1. The parenthesis is missing from line 155.

In addition a comma should be replaced with a dot. Therefore, it should be as following: ‘To overcome the lower HR modulation-stability Cottin et al. (2004) [18] proposed a’.
2. The dot is missing from the line 353. It should be: ‘Gorban et al. (2010) [67] studied the behavior of systems approaching a critical tran-‘.
3. The spaces is missing from line 484 for p values. It should  be: ‘0.91 95C I% (0.83-0.95) for HR (p < 0.001)’.
4. There is a non-uniform record, because once is 0.001 (for example on line 450) and once is .001 (for example on line 439). A uniform record is therefore necessary.

Authors: We have changed the text as suggested in note 1-4.

Reviewer 2 Report

The topic is promising, however, it is suggested to improve the following aspects:

1. To change the title by Detecting metabolic thresholds from nonlinear analysis of heart rate time series: A review

2. Abstract section must describe a metric of metabolic threshold as a result of the review

3. To increase the figures' resolution 

4. The conclusion section must describe clearly the contribution of the manuscript, in this sense, DFA and RQA should be described using metrics of metabolic threshold that in a concrete way explain in detail the benefits, advantages, limitations, applications and future works.

5. Authors mention the test to (normal is health?) female subject, for this reason, it is necessary to include the Informed Consent Statement.

6. In the conclusion section, authors mention in line 705: red and green line respectively in Figure 1, but figure 1 has no colors.

Author Response

Thank you for your kind message concerning our manuscript entitled “Detecting metabolic thresholds from nonlinear analysis of heart rate time series”. 

In the following you can find our answers

A:   and in italics if a phrase of the text is reported

REVIEWER 2

The topic is promising, however, it is suggested to improve the following aspects:

  1. To change the title by Detecting metabolic thresholds from nonlinear analysis of heart rate time series: A review

A: We have changed the titles as suggested

  1. Abstract section must describe a metric of metabolic threshold as a result of the review

A: We added in abstract:

 “While the advantages and disadvantages of each method, and the possible applications, are presented, this review confirms that the nonlinear analysis of HR time series represents a solid, robust and noninvasive approach to assess metabolic thresholds.”

  1. To increase the figures' resolution.

A: we uploaded figures, separately with higher resolution as possible.

  1. The conclusion section must describe clearly the contribution of the manuscript, in this sense, DFA and RQA should be described using metrics of metabolic threshold that in a concrete way explain in detail the benefits, advantages, limitations, applications and future works.

A: As new metric on respect to gas exchanged  and blood lactate, we proposed HR time series that are the data to monitor and analyze. On respect the analysis method we proposed RQA and the new description are DET and LAM

We specified as requested:

 at row 709, we added “respectively detected by DETmin and LAMmin,”

at Row 594 we named minimum of determinism and laminarity as (DETmin and LAMmin)

In section 5 titled “Know issues in using nonlinear methods” we have explained limitations and in conclusion the benefits and advantages (and at row 65-66).

As suggested in conclusion we explain in detail benefits advantages, applications and future works. At Row 705 we added

“The threshold points can also be quantitatively determined as they correspond to an abrupt change in determinism and laminarity, respectively detected by DETmin and LAMmin, in the epoch-by-epoch RQA procedure (see the Appendix A3). At both thresholds OLP regression analysis had very strong correlations between RQA and gold standard across HR, VO2 and workload.

In conclusion, the nonlinear analysis of HR time series can be considered a valid alternative metric to assess metabolic thresholds furnishing to medical and paramedical staff a easy-to-read and -use parameters. Furthermore, since HR data can be easily ac-quired by many low-cost devices, included smartwatches, it is reasonably expected a large widespread of nonlinear analysis of HR time series for a personalized evaluation of the fitness level and planning/monitoring of training or rehabilitation procedures.”

  1. Authors mention the test to (normal is health?) female subject, for this reason, it is necessary to include the Informed Consent Statement.

A: We changed normal in “healthy”, this female subject is one of the authors (GZ)

  1. In the conclusion section, authors mention in line 705: red and green line respectively in Figure 1, but figure 1 has no colors.

A: We apologize is related to Figure 5 (changed)
